# Class III Peroxidases in the Peach (*Prunus persica*): Genome-Wide Identification and Functional Analysis

**DOI:** 10.3390/plants13010127

**Published:** 2024-01-02

**Authors:** Ekaterina Vodiasova, Yakov Meger, Victoria Uppe, Valentina Tsiupka, Elina Chelebieva, Anatoly Smykov

**Affiliations:** 1Federal State Funded Institution of Science “The Labor Red Banner Order Nikita Botanical Gardens—National Scientific Center of the RAS”, Nikita, 298648 Yalta, Russia; megeryakov1@gmail.com (Y.M.); vikauppe@mail.ru (V.U.); valentina.brailko@yandex.ru (V.T.); elina.chelebieva@gmail.com (E.C.); selectfruit@yandex.ru (A.S.); 2A.O. Kovalevsky Institute of Biology of the Southern Seas of RAS, 299011 Sevastopol, Russia

**Keywords:** class III peroxidase, *Prunus persica*, peach, different tissues, cold stress, phylogeny, transcriptome analysis, qRT-PCR

## Abstract

Class III peroxidases are plant-specific and play a key role in the response to biotic and abiotic stresses, as well as in plant growth and development. In this study, we investigated 60 *POD* genes from *Prunus persica* based on genomic and transcriptomic data available in NCBI and analysed the expression of individual genes with qPCR. Peroxidase genes were clustered into five subgroups using the phylogenetic analysis. Their exon–intron structure and conserved motifs were analysed. Analysis of the transcriptomic data showed that the expression of *PpPOD* genes varied significantly in different tissues, at different developmental stages and under different stress treatments. All genes were divided into low- and high-expressed genes, and the most highly expressed genes were identified for individual tissues (*PpPOD12* and *PpPOD42* in flower buds and *PpPOD73*, *PpPOD12, PpPOD42*, and *PpPOD31* in fruits). The relationship between cold tolerance and the level of peroxidase expression was revealed. These studies were carried out for the first time in the peach and confirmed that chilling tolerance may be related to the specificity of antioxidant complex gene expression.

## 1. Introduction

Class III peroxidase (*POD* or *PRX*, EC number 1.11.1.7) is a protein superfamily that plays two important roles in plants: development and growth (1) and stress responses and resistance to abiotic and biotic factors (2) [1,2]. It is widely distributed in different organisms from bacteria to animals. Class III peroxidase only exists in plants and has a lot of functions—the removal of H_2_O_2_, the oxidation of toxic reductants, the biosynthesis and degradation of lignin, suberisation, auxin catabolism, and the response to environmental stresses such as wounding, cold, drought, pathogen attack, disease oxidative stress, etc. [3,4,5,6,7]. With the advent of genome-wide analysis, it has become possible to identify genes encoding peroxidases and analyse their structure. It has been shown that the number of genes varies in different plant species, and the evolution of this superfamily is inextricably linked to doublings and duplications. To date, the peroxidase (*POD*) family members have been characterised in several plants, including 73 *PODs* in *Arabidopsis* [8], 91 *PODs* in cassava [9], 90 *PODs* in *Betula pendula* [10], 138 *PODs* in *Oryza sativa japonica* [11], 93 *PODs* in *Populus trichocarpa* [12], 102 *PODs* in *Medicago sativa* [13], 94 *PODs* in *Pyrus bretschneideri* [14], 79 *PODs* in *Citrullus lanatus* [15], and 124 *PODs* in *Glycine max* [16].

Although *PODs* are known to play a key role in cell growth and the response to abiotic stress, the specific function of each family member is still unclear. Recently, much research has been devoted to the study of different classes of peroxidases and their specific roles. It was revealed that under low-temperature conditions, some *BpPODs* showed different expression patterns at different times in *B. pendula* [10]. In the carrot, some genes encoding peroxidases (in particular, *DcPrx23*, *DcPrx30*, *DcPrx32*, and *DcPrx62*) have been shown to probably have a specific function and are involved in the flavonoid/anthocyanin or lignin pathway [17]. In addition to the specific functions of the different members of the peroxidase family, there are peculiarities of gene expression in tissues. For example, in cassava, the *MePOD23* gene was induced after H_2_O_2_ treatment in leaves but showed an opposite trend of expression in storage roots, indicating their differential role in different tissues [9].

Since different peroxidase genes can exhibit specific functions (as discussed above), it is important to analyse their expression patterns under different stresses. The data obtained can be used for genetic engineering to create plant cultures with increased resistance to various factors. The first works in this direction have already shown the promise of such studies and the need to study the functional specificity of all members of the peroxidase family. For example, transgenic wheat plants with the overexpression of one gene encoding *POD* (*TaPRX-2A*) were created. When compared with the wild type (WT), they showed higher resistance to salt stress [18]. In soybeans, 23 genes showing differential expression between drought-tolerant and drought-sensitive genotypes under stress were identified based on transcriptomic data. With one of these genes (*GsPOD40*), transgenic soybean lines showing constitutive gene over-expression were obtained. This line showed significantly higher drought tolerance compared with wild-type (WT) plants under drought stress [16].

The peach (*Prunus persica* L.) is one of the main stone fruit crops in the world. It is grown in more than 70 countries. Plantations occupy over 1.5 million hectares. Over 20 years, the world production of peach fruits increased from 11.4 million tons in 1998 to 24.7 million tons in 2017. The peach is characterised by early maturity, high yield, and dessert qualities of fruits that contain many microelements and biologically active substances that have therapeutic and prophylactic significance for human health.

Since the peach is a commercially important fruit crop, studies on the peroxidase enzyme have been carried out for a long time: the first studies appeared in the 1970s in the 20th century [19]. The main part of the work is devoted to the analysis of peroxidase activity under various influences. The role of peroxidases in the lignification process [20], enzymatic browning reactions [21], and ageing of fruits after harvest [22] have been shown. Peroxidase activity varies between rootstock and may be a biochemical indicator of stress [23]. Also, peroxidase activity, together with other genes of the antioxidant complex, increases with increasing duration of stress under both mild and severe drought stress [24]. That said, there are very few studies examining peroxidase expression in peaches. A study examining the mechanisms of cold tolerance in two peach cultivars differing in cold tolerance identified two peroxidase genes whose expression increased under cold stress in the tolerant cultivar [25].

However, a full-genome analysis of the peroxidase family has not been performed so far. The existing peach assortment needs to be improved, as it is necessary to increase the adaptability of cultivars to winter frosts, spring frosts, and drought, as well as to fungal and viral diseases. Studying the structure and function of *POD* genes in *P. persica* is very important for understanding the growth and development process and the molecular mechanism of stress tolerance, and the results can provide a theoretical basis for genetic improvement in *P. persica*. So, this work aimed to study peach peroxidases based on genomic and transcriptomic data, describe the structure of genes encoding *PODs*, analyse their phylogeny, and study the expression of several genes in different cultivars and tissues with qPCR methods.

## 2. Results

### 2.1. Peroxidase Genes in P. persica

The analysis of the Lovell cultivar genome (GCA_000346465.2) showed that the peach genome contains 60 annotated genes encoding *POD*. All peroxidase genes revealed high nucleotide and amino acid variability. The protein length of *PpPODs* varied from 264 to 413 amino acids with an average of 332. The exon–intron structure also varied greatly: some genes are intronless, while others have several long introns. The number of exons varied from one to five, and most peroxidases had four exons (Figure 1a). Some of the genes were intronless: *PpPOD31* (position on the third chromosome: NC_034011:25,556,678–25,557,658) and two *PpPOD41* genes (first chromosome: NC_034009.1:26,947,434–26,949,244 and fifth chromosome: NC_034013:15,058,819–15,059,799). The maximum number of exons was in the gene *PpPOD* (sixth chromosome: NC_034014: 23,882,739–23,885,446).

All 60 *PpPOD* genes were widely distributed over eight chromosomes (Figure 1a). Both chromosome 5 and chromosome 8 contained the least number of *PpPOD* genes with only three. Chromosome 6 had the widest distribution with 19 genes followed by 13 on chromosome 1. In the remaining chromosomes, the number of *PpPOD* genes varied from 2 to 10. Moreover, all chromosomes as well as their regions had a different density of *PpPODs*. The highest density of *PpPOD* genes was observed at the top end of chromosome 6.

To investigate the evolutionary relationship of *PpPODs*, a phylogenetic tree was constructed with the 60 amino acid sequences of *P. persica* and 73 of *A. thaliana* (Figure 2). The phylogeny showed high diversity among the peroxidase gene family even inside the subgroups. The overall pairwise distance was 1,02 for amino acids and 0,98 for nucleotides. Some *PODs* were represented in variable isoforms and had more than one copy in the genome. Based on the phylogenetic analysis, all *PpPOD* genes were divided into six distinct subgroups from A–F. The number of *PpPODs* in the subgroups varied from 5 to 19. Subgroup E with 19 *PpPOD* members was the largest subgroup for peach. It was followed by A (12), C (9) and D (9). The small subgroups B and F contain five and six *PpPOD* members, respectively.

The motif distribution within the *POD* gene family was determined for *A. thaliana* and *P. persica*, and 15 motifs were identified (*p* < 0.0001) (Figure 2). A total of 20 variants with a conserved motif distribution were obtained for *P. persica*. Most peroxidase genes have the same structure in the central part and greater variability at the beginning and end of the gene. Most *PpPODs* are characterised by the following sequence of motifs in the middle of the gene: 6-2-9-10-5. In addition, there is a conserved region at the beginning (motifs 7-1) and at the end of the genes (motif 4, and in rare cases, without it). Of the 15 motifs, 6 are characteristic of the peroxidase family. Based on InterProScan data, motifs 1, 2, 3, 3, 4, 5, and 12 belong to the haem peroxidase superfamily (IPR010255). In addition, motif 1 is a peroxidase active site (IPR019794) and motif 12 is a peroxidase haem–ligand binding site (IPR019793).

Analysis of conserved motifs in peach revealed some relationships between their distribution and clustering. The number of conserved motif sequence variants varies in different clades. Sequence variability in the conserved motifs ranges from 0.44 (subgroup C—four variants for nine genes in peach) to 0.8 (subgroup B—four variants for five genes in peach). Some of the conservative motifs and their distribution characterised certain subgroups. Thus, motif 14 is found only in subgroup E. Some variants of motif distribution are found only in one subgroup (for example, 1, 2, 5, and 8 in subgroup E). Subgroups B and C are distinguished by conservative motif sequences (4, 7, 12, 20), which are not repeated anywhere else. The most common sequence of conservative motifs is 13, which is included simultaneously in four subgroups of peroxidases (A, C, E, and F).

### 2.2. Expression of Peroxidases in Different Cultivars

The expression levels of *POD* genes in different peach cultivars were investigated using RNA-Seq analysis on the flower bud tissue of five genotypes of *P. persica* available in NCBI (Figure 3). The expression of *POD* showed huge variability across cultivars. Clustering identified two clusters of genes (A and B), which can be designated as low- and high-expressed. All calculated values of FPKM and DeSEQ2-normalised counts are represented in Appendix A. Gene expression levels between the biological samples for the same cultivar do not differ much, in contrast to the expression level in different peach genotypes (Figure 3a,b). The subset of genes that also showed high expression is highlighted in cluster B-II (*PpPOD31*, *PpPOD44*, *PpPOD4*, and *PpPODP7)* and B-I (*PpPOD47*, *PpPOD18*, *PpPODA2*, *PpPOD4*, *PpPOD11*, *PpPODA2*, *PpPOD73*, *PpPOD16*, and *PpPOD17*). These genes belong to subgroups A, C, D, and E. Cluster A represents peroxidase genes with low values. Cluster A includes all genes of subgroup B and most genes of subgroups C and F. The total expression calculated as DeSEQ2-normalised counts for each cultivar ranged from 9106 to 11639. Analysis revealed a possible positive correlation between total peroxidase gene expression levels and cold tolerance based on chilling unit (CR) data (r^2^ = 0.77) (Figure 3b). The most highly expressed genes are *PpPOD12* and *PpPOD42*, are also differ in genotypes with various tolerance to cold: gene *PpPOD12* expression increases in more cold tolerant genotypes, while gene *PpPOD42* expression decreases.

To investigate the relative expression level of *PpPODs* in different tissues (plant shoot, leaf, and fruit), five non-randomly selected genes with different motif structures were tested with qPCR. The genes *PpPOD4*, *PpPOD15*, and *PpPODA2* belong to the most extensive subgroup E, *PpPOD5* to subgroup C and *PpPOD31* to subgroup A (Figure 2). Differences in the relative expression of peroxidases in tissues were detected in cultivars with different cold tolerance (“Asmik”—winter-hardy cultivar; “Springold”—weak resistance to spring frosts) (Figure 4).

The expression patterns in the cultivars varied for different tissues. The greatest difference was found for fruit, where all genes showed higher relative expression in the cold-tolerant cultivar “Asmik”. In the leaves, this difference was observed for two genes *PpPOD31* and *PpPOD5*. In the shoots, no differences in the relative expression between these cultivars were observed.

### 2.3. The Expression of Peroxidases under Stress Factors

The expression of *POD* genes under fungus infection (*Monilinia laxa*) was investigated with RNA-Seq analysis on the fruit tissue of *P. persica* available in NCBI (Figure 5).

The response to infection in immature and mature fruits differs; however, there are common patterns. All peroxidase genes were categorised into two groups: the low expression level and the high expression level (Figure 5a). The first group includes all genes of subgroup B and most genes of subgroup C. The high-expressed gene is divided into three groups. *PpPOD42*, *PpPOD73*, *PpPOD12*, and *PpPOD31* have the highest expression level in all samples and determine the total level of peroxidase expression in fruit (peroxidase subgroups F, A, D, and A, respectively). As a result of infection, the expression of some genes increased while others decreased in both groups of peaches (immature and mature fruit). The differently expressed genes (DEGs) were detected with DeSEQ2 (comparison between control and cold stress, Pajv < 0.01). The full results of the differential expression analysis are presented in Appendix A. The expression of the *PpPOD73, PpPOD44, PpPOD16, PpPOD15, PpPODP7, PpPOD4*, and *PpPOD12* genes increased at infection onset, while *PpPODA2*, *PpPOD17, PpPOD42*, and *PpPOD31* decreased both in the immature and mature fruits.

The pattern of total peroxidase expression differs between immature and mature fruits (Figure 5b). In immature fruits, the total peroxidase expression is higher in healthy samples than in fruits inoculated with fungus. In mature fruits, on the contrary, the total expression level of peroxidases increases two-fold 48 h after infection.

The analysis of peroxidase gene expression under cold stress revealed some common patterns in the control samples compared to expression patterns in the different cultivars (Figure 6). Genes that have either a low expression level or are not expressed are identified in a separate cluster. This cluster is represented mainly by peroxidase genes of subgroups B and C. Among the highly expressed genes, three clusters can be identified similar to the study of expression in different cultivars: cluster III (*PpPOD12* and *PpPOD42*), cluster II (*PpPOD31*, *PpPOD44*, *PpPOD4*, *PpPOD47*, and *PpPODP7)* and B-I (*PpPOD18*, *PpPODA2*, *PpPOD4*, *PpPOD11*, *PpPODA2*, *PpPODA2*, *PpPOD73*, *PpPOD16*, *PpPOD16*, and *PpPOD17*).

As in the case of infection, under cold exposure, the expression of some genes increased while others decreased even at low values of chilling units. The total expression of peroxidases increased at the onset, but with severe exposure to cold, the total expression started to drop significantly compared with the control. The full results of the differential expression analysis are presented in Appendix A. A significant increase in expression for some genes was detected when crossing the threshold value of cold stress: for the cultivar “Fantasia”, it was 500 CU (Padj = 2.1 × 10^−12^). Such an increase in expression was found for the following genes: *PpPOD16* and *PpPOD16* (subgroup A); *PpPOD41* (subgroup C); *PpPOD9* and *PpPOD40* (subgroup D); and *PpPOD4* (subgroup E). The down-regulated genes were *PpPOD47* (subgroup A); *PpPOD44* (subgroup C); *PpPOD12* (subgroup D); *PpPODA2 (1)* and *PpPODA2 (2)* (subgroup E); and *PpPOD42* (subgroup C). The *POD* expression of subgroup B genes was not changed with cold stress. Genes that were not expressed before also began to be expressed.

## 3. Discussion

Class III peroxidase is a plant-specific enzyme that plays a crucial part in responding to biotic and abiotic stress, as well as regulating plant growth and development. It is coded in organisms by a large family of genes, with some species having over a hundred duplications. Various genes have functional specificity. The Class III peroxidase gene family of *Arabidopsis* [8], *O. sativa japonica* [11], *P. trichocarpa* [12], *M. sativa* [13], maize [26], *P. bretschneideri* [14], cassava [9], *Brachypodium distachyon* [27], *Solanum tuberosum* [28], *B. pendula* [10], *C. lanatus* [15], *G. max* [16], *Gossypium hirsutum*, *G. arboretum*, *G. raimondii* [29], *Nicotiana tabacum* [30], *Capsicum annuum* [31], and sugarcane [32] have been researched; however, no such investigations have been carried out on the peach.

In this study, the gene structure and expression of 60 peroxidase genes in the peach genome were analysed. Comparative phylogenetic analyses revealed significant diversity among *PpPODs* in the peach. *PpPOD* genes were classified into six distinct groups (A-F), which is consistent with previous reports in cassava [9] and soybean [16]. The *PpPOD12* and *Arabidopsis* (AT1G71695) *POD* sequences form a separate branch, although their inclusion in clade D is uncertain. Subgroup F displayed a comparable phenomenon, with the peach and *Arabidopsis PODs* included even though they have a high genetic distance. This observation, similarly, holds for *Arabidopsis*, where Tognolli partitioned the 73 *AtPrxs* into five main groups (Gr1-Gr5) alongside two single-member branches (*AtPrx12* and *AtPrx48*) [8]. Further research by Meng on peroxidases in the carrot also created two distinct subgroups, grouping *AtPrx12* and *AtPrx48* with other genes separately [17]. While our investigation indicates that certain genes might be single-member branches, this could be due to clustering features with a limited dataset.

The exon–intron structure of peroxidase genes exhibits differences that are not associated with phylogenetic relatedness or motif distribution. Such variation in gene structure may be crucial in understanding the evolution and function of gene superfamilies [33,34]. The number of introns is generally low, with most genes containing four introns; this figure is comparable to the average value of other plants, such as the soybean [16], which has three, and *B. pendula* [10], which has six. It is known that stress and rapidly regulated genes are deficient in introns, and this explains the low number of introns in peach peroxidase genes [35]. The necessity for prompt reaction to adverse conditions and increased expression of PODs also clarifies the widespread dispensation of coding genes across all chromosomes, a trend that is also observed in other plants [8,9,11,12,14,26].

Fifteen conserved motifs were identified during this study. While the motifs are represented differently in various genes, their distribution remains conservative. The six motifs (1, 2, 3, 4, 5, 12) are characterised as a peroxidase superfamily, two of which consist of a peroxidase active site (motif 1) and a peroxidase haem–ligand binding site (motif 5). The genes *PpPOD5*, *PpPOD7*, *PpPOD11*, *PpPOD15*, *PpPOD17*, *PpPOD24*, *PpPOD64*, and *PpPOD66* do not contain some of the six motifs and are generally low-expressed. However, the *PpPOD11* gene has average expression values and is part of a cluster of highly expressed genes in flower buds (refer to Figure 3 and Figure 6). This gene is without motif 4, which comprises 21 amino acids and is not an active or haem–ligand binding site. *PpPOD17*, *PpPOD64*, and *PpPOD66* also exhibit mean expression levels across varied phenotypes, tissues, or exigencies in spite of the absence of motif 4. Conversely, *PpPOD5*, *PpPOD7*, and *PpPOD15* are noticeably under-expressed, presumably because of the concurrent absence of multiple peroxidase domain motifs. It is noteworthy that motif 14 is only found in *PpPODP7* and *PpPOD4*, coinciding with the highly expressed values of these genes in all scrutinised transcriptomes. This motif is present in Arabidopsis and is likely associated with the specific function of the peroxidase protein. Among all the transcriptomes, the genes *PpPOD12* and *PpPOD42* exhibit the highest levels (the 14th variant of conserved motifs distribution). Although the *PpPOD21*, *PpPOD29*, *PpPOD40*, and *PpPOD47* genes have the same motif set, their expression levels are lower. So, the results imply that there is no direct correlation between gene expression and the composition of conserved motifs.

At the same time, it was established that several peroxidase genes underwent negative selection [10], which suggests the potential for distinct functions of various genes within this family in relation to their structure and amino acid sequence. Moreover, previous studies have shown that different peroxidase genes exhibit specific responses to stress factors at different developmental stages or in various tissue types, potentially indicating specific gene functions [29]. Thus, the *TaPRX-2A* gene in *T. aestivum* has a positive function in reacting to salt stress [18], and *TaPRX111*, *TaPRX112*, and *TaPRX113* were found to respond to nematode infection [36]. In *Z. mays*, five POD genes (*PRXs ZmPRX26*, *ZmPRX42*, *ZmPRX71*, *ZmPRX75*, and *ZmPRX78*) are involved in the response to various abiotic stress [26]; in *G. max*, the overexpression (OE) lines of GsPOD40 showed considerably higher drought tolerance compared with wild type (WT) plants under stress treatment [16]; and in *C. annuum*, two POD genes are upregulated and seven are downregulated during fruit ripening [31]. *Arabidopsis* is the most extensively researched plant model. Its response to wounding and fungal stresses involves the genes *AtPRX21*, *AtPRX62*, and *AtPRX71* [37,38]. Furthermore, cell elongation is associated with *AtPRX33* and *AtPRX34* [39], and lignification is associated with *AtPRX72* [40].

In our research, we identified specific peroxidase expression patterns by analysing experimental and transcriptome data available in NCBI. We investigated five different cultivars and hybrids (flower buds) and the effect of fungal infection (fruit) and cold stress (flower buds) using RNA-Seq data. All peroxidase genes within the samples were categorised into high- and low-expressing genes. The composition of these groups was nearly identical between varieties and their hybrids, but differences were observed between different tissues (flower buds and fruits). Highly co-expressed genes such as *PpPOD17*, *PpPODP7*, *PpPOD47*, *PpPOD4*, *PpPOD44*, *PpPODA2*, *PpPOD73*, *PpPOD31*, *PpPOD42*, and *PpPOD12* were detected. *PpPOD12* and *PpPOD42* are the most highly expressed genes in flower buds, whereas in the fruit, these genes are *PpPOD12* and *PpPOD73*. In all the samples examined, all subgroup B genes and the majority of subgroup C and F genes were part of the low-expressed gene cluster, possibly due to their primary structure. Using *PpPODA2*, *PpPOD4*, *PpPOD5*, *PpPOD15*, and *PpPOD31* as examples, differences in gene expression across fruits, leaves, and shoots were validated with RT-PCR. The findings are consistent with the results of other studies [10,41]. Tissue-specific peroxidase expression is attributed to its significant role in plant development.

Significant differences were found in the same tissue type but at different developmental stages (Figure 5). The overall expression level of peroxidases is higher in immature fruit than in mature fruit. This may account for the greater resistance of immature fruit to infection [42]. Similar results were previously obtained for pear peroxidases, the expression of which was significantly correlated with the lignin content during fruit development [16]. The response to fungal infection also varies in different fruit stages. In the case of *M. laxa* infection in immature fruits, the expression of peroxidases decreases, whereas in mature fruits 48 h after infection, it sharply increases.

This study also investigated the effect of cold stress on peroxidase expression and identified patterns between cold-tolerant and cold-sensitive cultivars. Low temperature is a critical limiting factor for plant growth, development, and reproduction. Stress is usually associated with the accumulation of reactive oxygen species (ROS), including hydrogen peroxide (H_2_O_2_), hydroxyl radicals (OH-), and superoxide radicals (O-) in plant cells. High levels of ROS lead to lipid peroxidation and membrane damage in plants under cold stress [43,44]. Antioxidant enzymes play a critical role in ROS scavenging and influence cellular ROS levels [45,46]. Peroxidase is one of the most important antioxidant enzymes, and its activity increases in different cultivars under cold stress [47].

Our results demonstrated an increase in the total expression level of peroxidase genes during cold exposure of peach flower buds at the initial stage of exposure and then a sharp drop in the expression level when passing through the CR threshold value (Figure 6). Furthermore, the response to short-term chilling stress shows that after the tolerance period (for “Fantazia”—500 chilling units) there was a significant change in the expression of some genes, indicating an active counteraction to stress.

A correlation between the peroxidase expression pattern and varieties with different cold tolerance was also found. The different expression levels in “Cold Princess” and “21st Century” cultivars with different tolerances were mentioned earlier [25]. Two peroxidase genes (*PpPOD31* and *PpPODA2*) were up-regulated in the cold-tolerant cultivar under cold stress but remained unchanged in the cold-sensitive cultivar based on RT-PCR. In our study, these genes were not up-regulated under cold stress in the “Fantasia” cultivar. So, an analysis of the expression of only 2 out of 60 peroxidase genes is not sufficient due to the possible diversity of functional specificity. Our analysis, based on transcriptomic data, showed that no single gene expression pattern was found for the different cultivars, but the total gene expression revealed a positive correlation between the resistance of peach genotypes and the level of total expression (Figure 3b). Total peroxidase gene expression increased with increasing resistance. The transcriptome analysis agreed with the real-time results: some genes showed opposite expression patterns in cold-tolerant and cold-sensitive peach cultivars and significantly higher expression levels in the cold-tolerant cultivar “Asmik” (Figure 4, fruits, leaves). Previously, other researchers hypothesised that the main difference between cold-tolerant and warm-loving plants is that in the former, the formation of reactive oxygen species during chilling does not lead to oxidative stress. This suggests that cold tolerance may be related to the increased level of antioxidant complex gene expression.

## 4. Materials and Methods

### 4.1. Description and Phylogeny of Peroxidases in P. persica

Ten genomic assemblies of different quality are present today for different peach cultivars in the National Center for Biotechnology Information (NCBI): 5 assemblies have a chromosomal level (GCA_000346465.2, GCA_024337555.1, GCA_015730445.1, GCA_018340835.1, GCA_022343065.2), 4 have a scaffold level (GCA_020226405.1, GCA_000218175.1, GCA_000218215.1, GCA_000218195.1) and 1 has a contig level (GCA_019209885.1). For this investigation, we analysed the genome assembly of *P. persica* cultivar “Lovell” (GCA_000346465.2), which is the reference for the species. Coding sequences and protein sequences of all PODs were retrieved from the NCBI RefSeq database (GCF_000346465). We used the protein sequence of the 73 Arabidopsis as the query (http://www.arabidopsis.org/index.jsp, accessed on 1 July 2023). The locations of the peach *POD* genes were determined by analysing their chromosomal distribution using Genome Data Viewer available in NCBI for this assembly. The exon–intron structure of each POD was described according to the annotation of *P. persica* genome.

A phylogenetic analysis was conducted using MEGAX software, Version 10.2.6 (https://www.megasoftware.net/, accessed on 21 October 2022) [48,49]. The dataset consisted of all *POD* protein sequences of *P. persica* and *A. thaliana*. Multiple alignment of deduced amino acid sequences was performed using Muscle [50]. The phylogenetic tree was constructed with the Maximum Likelihood (ML) method and the Whelan and Goldman model [51]. The consistency of the ML tree was validated by setting a bootstrap value of 1000. The final phylogenetic tree was visualised with MEGAX. A motif analysis was performed by MEME online tool on Classic mode: (https://meme-suite.org, accessed on 21 October 2022). Fifteen motifs were found between 6 and 200 wide for the set of all 131 protein sequences between 71 and 413 in length (average length: 327.3).

### 4.2. Expression of Some Peroxidases Based on RT-PCR

#### 4.2.1. Sample Collection

Two different peach cultivars, “Springold” and “Asmik”, were selected to study peroxidase expression in different cultivars and tissues. They are grown in the collection of the Nikita Botanical Gardens, located on the southern coast of Crimea, near the city of Yalta. The plants were planted in the collection in 2010 according to the planting scheme 5 × 3 m, on an almond rootstock with a drip irrigation system.

For “Asmik”, seedlings of the “Chughuri” cultivar were bred by the Armenian Research Institute of Viticulture, Winemaking and Fruit Growing. The tree is medium-sized, with an outstretched paniculate crown. At the time of fruiting, it enters the third year after planting, and the yield is plentiful and regular. The flowers are rose-shaped, and the glands are oval. The fruits are medium-sized (145 g), rounded, and greenish cream with a dark red blush by 25%. The cultivar is not picky about growing conditions, is winter-hardy, and is weakly affected by fungal diseases [52,53,54,55].

For “Springold”, the cultivar was bred in the USA as a result of a complex cross [(Fireglow × Hiley) × Fireglow] × Springtime. The tree is medium-sized, with a wide-obverse-cone-shaped crown of medium density. The flowers are rose-shaped, and the glands are oval. The fruits are medium (100–120 g), rounded, and sometimes slightly elongated or narrowed towards the top. The base is rounded. The ventral suture is weak and slightly cracked. Winter hardiness and resistance to spring frosts are weak [55,56,57,58,59,60,61,62,63].

In addition to these cultivars, this study included a bioinformatic analysis of transcriptomes of cultivars available in NCBI: “Fantasia” [64,65,66,67,68], “Venus” [69,70,71], “Cold Princess”, and “21th Century” [72]. A description of all the cultivars analysed in this study is given in Appendix A.

#### 4.2.2. Molecular Analyses

Total RNA was extracted from plant shoots, fruits, and leaf samples using an innuPREP RNA Mini Kit (Analytik Jena, Jena, Germany). RNA purity and integrity were assessed using the A_260/A280_ absorbance ratio and stained with ethidium bromide in a 1.5% agarose gel. cDNA was synthesised from 500 ng of purified total RNA using MMLV Reverse Transcriptase (Evrogen, Moscow, Russia) following the manufacturer’s protocol.

The cDNA synthesised was used as a template. Quantitative real-time PCR analysis (RT-qPCR) was performed on a LightCycler^®^ 96 Instrument (Roche, Basel, Switzerland) using a qPCRmix-HS kit with SYBR GreenI (Evrogen, Moscow, Russia). The reaction mixture (total volume 15 μL) contained 1 μL of cDNA and 0.4 μM of each primer. The reaction conditions consisted of an initial denaturation at 95 °C for 3 min, followed by 50 cycles of 95 °C for 10 s, 60 °C for 10 s, and 72 °C for 15 s. Melting curve data were collected at 65–95 °C (0.5 °C/s). All reactions were performed with three replicates. For each run, a negative control without a template was included.

*β-actin* (*ACT*) and glyceraldehydes-3-phosphate dehydrogenase (*GAPDH*) were used as reference genes for real-time PCR analyses [52]. Various genes encoding peroxidases were selected for expression studies. Some were used previously during different physiology investigations; other genes were explored for the first time. All gene-specific primer sequences are provided in Table 1.

The data analysis was carried out using Roche software. Efficiencies of amplifications were determined by running a standard curve with serial dilutions of cDNA. For each measurement, a threshold cycle value (Cq) was determined as the fractional cycle number at which the fluorescence passes the fixed threshold. The descriptive statistics of the expression levels were computed for each candidate reference gene using the software package BestKeeper [74]. The relative expression levels of each target gene were normalised by calculating the geometric mean of the *ACT* and *GAPDH* genes using the ΔΔCt comparative method [75]. As the stability of these genes in different tissues was poor, the comparison between cultivars in one tissue was provided (average length 327.3).

### 4.3. Expression of Peroxidases Based on Transcriptomic Data

#### 4.3.1. Different Cultivars

To analyse the expression of the whole *POD* family, a transcriptome analysis of the available RNA-seq data in NCBI was performed. In order to avoid errors associated with different sequencing approaches, cDNA sequencing libraries for Illumina were selected. Since sequencing from different studies was used, the essential task for correctly analysing gene expression was the grouping of data by individual traits. The libraries chosen for comparative analysis were divided into two groups—various cultivars of *P. persica* and stress sets. Since the expression of genes of this family may differ in tissues (leaf, bud, shoot, fruit), libraries derived from the same tissues were selected for comparative analysis of POD expression in different cultivars. The highest representation of cultivars and hybrids was for flower buds (“Fantasia”, peach F2 hybrids A209, A340, A318, and A323 derived from cultivar “Fla.92-2C” with “Contender”). These cultivars are characterised by different cold tolerance: “Fantasia”—750 chilling requirements (CR), genotype A209—300 CR, genotype A340—300 CR, genotype A318—850 CR, and genotype A323—1100 CR [76,77,78]. A total of 17 SRAs in this category were selected (NCBI accession numbers SRR10269838-SRR10269852, SRR17074239, and SRR17074240). A more detailed description of the transcriptomis data analysed is given in Appendix A.

#### 4.3.2. Stress Conditions

The effect of stress on the level of peroxidase expression was also studied. For this purpose, experimental data available in NCBI were analysed. The first analysed experiment (BioProject: PRJNA610066), was dedicated to the expression of this gene family upon infection with the fungus *M. laxa* in the mature and immature fruits of *P. persica* cultivar “Venus” [42]. The immature and mature fruit were resistant and susceptible to brown rot, respectively. The gene expression of healthy fruit and after *M. laxa* infection were compared. RNA-seq data was at 6, 14, 24, and 48 h post-inoculation (hpi). The second analysed experiment was dedicated to study the effect of cold stress on peroxidase gene expression in the peach, and the PRJNA784945 dataset was used to compare the effects of cold on flower buds [79]. Flower buds were collected at four time points, corresponding to 0, 200, 475, and 770 chilling units (CUs), during the autumn/winter vegetative arrest season.

#### 4.3.3. Bioinformatic Analyses

All transcriptomes were analysed using the same pipeline: quality assessment, filtering and trimming, mapping to the peach peroxidase database (based on the reference genome assembly of *P. persica* cultivar “Lovell” GCA_000346465.2), and calculating the scores to estimate the expression level of each *POD* gene. Quality and length trimming of the reads were conducted using fastp v.0.23.2 [80]. Raw reads with a length of less than 80 bp and a quality lower than Q20 were excluded from the analysis as well as reads containing adapters and poly-N. Mapping to the peach peroxidase database was carried out using a pseudo-alignment algorithm implemented in Kallisto v0.48.0 [81]. FPKM (fragments per kilobase of exon per million fragments mapped) was used for the comparative estimation of *POD* gene expression in each peach cultivar and was calculated using a simple Python script from the rnanorm package (https://github.com/genialis/RNAnorm, accessed on 10 September 2023). In this case, we needed to reveal the relative expression between different peroxidase genes. This task was realised using Kallisto v.0.48. For the SE library, the average fragment length was set at 150, for PE, all parameters were set at default with automatic average fragment length calculation. Normalised counts were used for gene count comparisons between samples to understand the gene expression changes under stress conditions (infection by the fungus *M. laxa* and cold stress). This approach was realised using DESeq2 [82]. It was chosen because DeSEQ is able to detect outliers and excludes genes with extreme read counts by default, and the false positive rate for DEG is 0% at adjusted *p*-values less than 0.05 [83,84]. The expression levels were visualised with a heatmap in R. The total expression of all encoding peroxidase genes was calculated separately by summing raw counts and then recalculating using a specific normalised method. All results were stated as mean ± standard error (SE). A Mann–Whitney test was used to analyse the difference between groups. Data analyses (graphics and statistics) were accomplished using R.

## 5. Conclusions

Peach peroxidase genes form a diverse family with robust clustering, distinct exon-intron structure, and limited variation in the composition of conserved motifs. Despite some patterns, no significant association between gene structure (phylogeny, number of introns, composition of conserved motifs) and expression level was established. Peroxidase genes are clustered into low-expressed and high-expressed genes. A notable difference in gene expression was detected in various tissues. When stress is initiated, the overall expression of peroxidases elevates but then drops as it crosses the tolerance threshold. The distinct genes exhibit increased and decreased expression depending on the type of stress. Our study demonstrated a correlation between the overall peroxidase expression and the cold tolerance of different varieties. Therefore, the expression pattern of *POD* may indicate the stress tolerance of cultivars of peach. The DEGs identified in this study when exposed to cold and contamination may be promising for bioengineering, which requires further studies.

## Figures and Tables

**Figure 1 plants-13-00127-f001:**
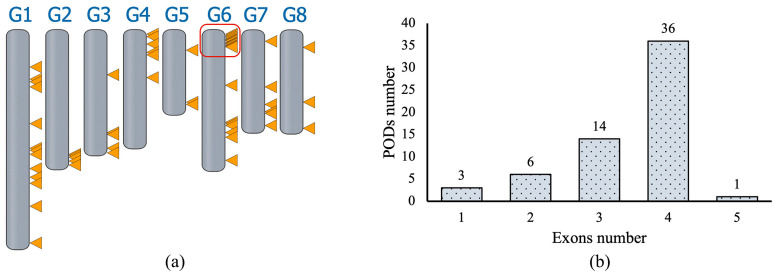
Localisation and exon number of peroxidase genes in *P. persica*. (**a**) Chromosomal location of 60 *POD* genes on the eight peach chromosomes. The red box shows the region with the highest density of *PpPOD* genes. (**b**) The distribution of *POD* genes with different numbers of exons. The region with the highest density of *PpPODs* is marked with a red box.

**Figure 2 plants-13-00127-f002:**
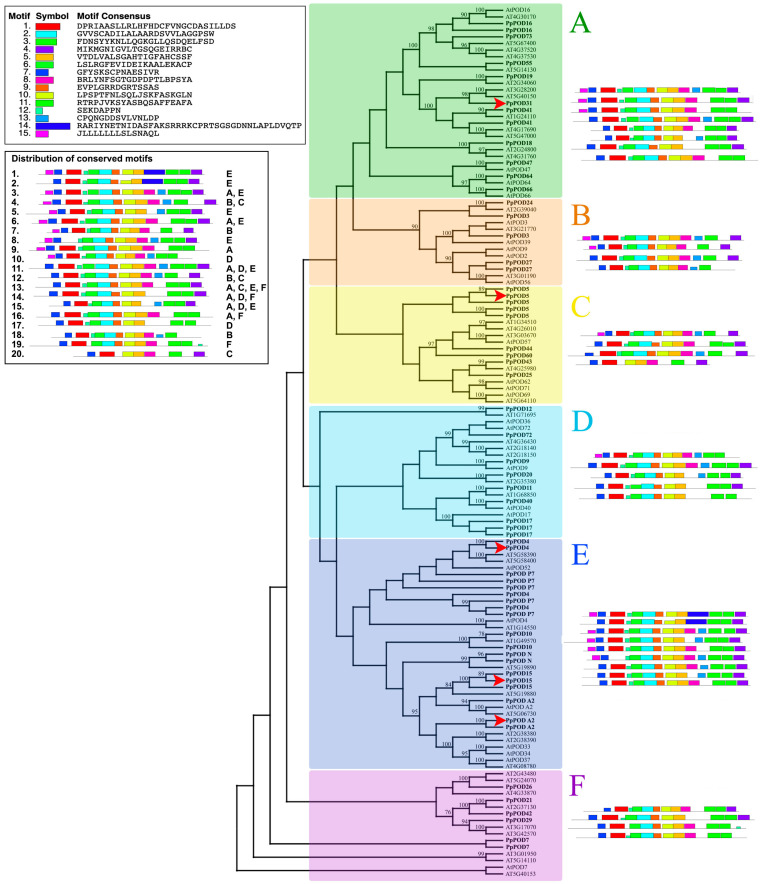
The evolutionary relationship and the motif analyses of *POD* family members in *P. persica* and *A. thaliana*. The unrooted phylogenetic tree on the left was constructed with the Maximum Likelihood (ML) framework with the WAG model using the MEGA X program. The different colours and letters (**A**–**F**) indicate the clade representing six phylogenetic subgroups. The sequences from *P. persica* were marked as *PpPOD*. Red arrows point to genes whose expression has been studied by qPCR in different tissues. The 15 different colours of the boxes on the left represent diverse conserved motifs, while the grey lines indicate non-conserved sequences. The conserved motifs were identified with the MEME database. All variants of conservative motif sequences for *P. persica* are presented on the left and for each clade.

**Figure 3 plants-13-00127-f003:**
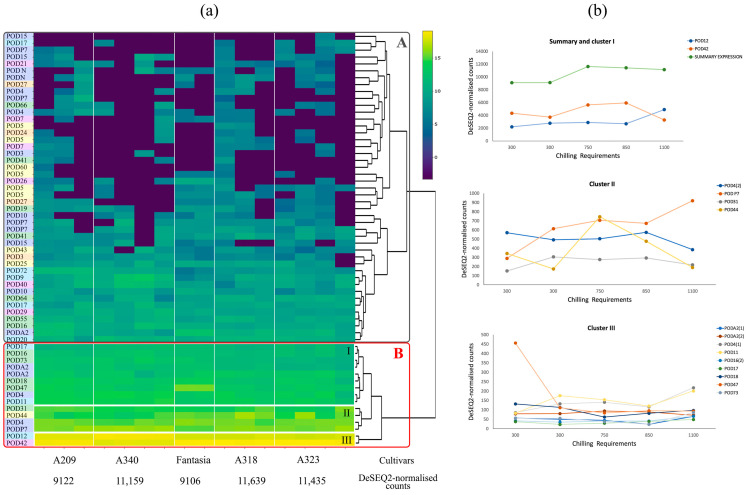
Expression pattern of the *POD* genes in different peach cultivars based on the transcriptomic analyses (flower buds). A209, A340, A318, and A323 are F2 hybrids derived from cultivars “Fla.92-2C” with “Contender”. (**a**) The heatmap of *POD* expression in five genotypes of the peach. The log2-based FPKM value was applied to build the heatmap. *POD* genes from subgroup A are marked with green boxes, subgroup B with orange boxes, subgroup C with yellow boxes, subgroup D with light blue boxes, subgroup E with purple boxes, and subgroup F with pink boxes. Bottom line—the total expression of all genes encoding peroxidase calculated as DeSEQ2-normalised counts. A and B denote low-expressed and high-expressed gene groups, respectively. I, II, III indicate three clusters in high-expressed gene group B. (**b**) The expression levels of *POD* genes from cluster B in cultivars with different tolerance to cold.

**Figure 4 plants-13-00127-f004:**
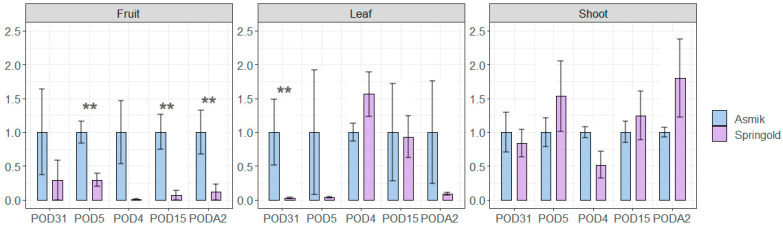
qRT-PCR analyses of *POD* genes in different tissues of *P. persica*. The Y-axis represents the relative expression level. The gene expression in “Springold” compared to “Asmik”. Data are presented as mean ± SE. The double asterisk on the bar represents a significant difference at ** *p* < 0.001.

**Figure 5 plants-13-00127-f005:**
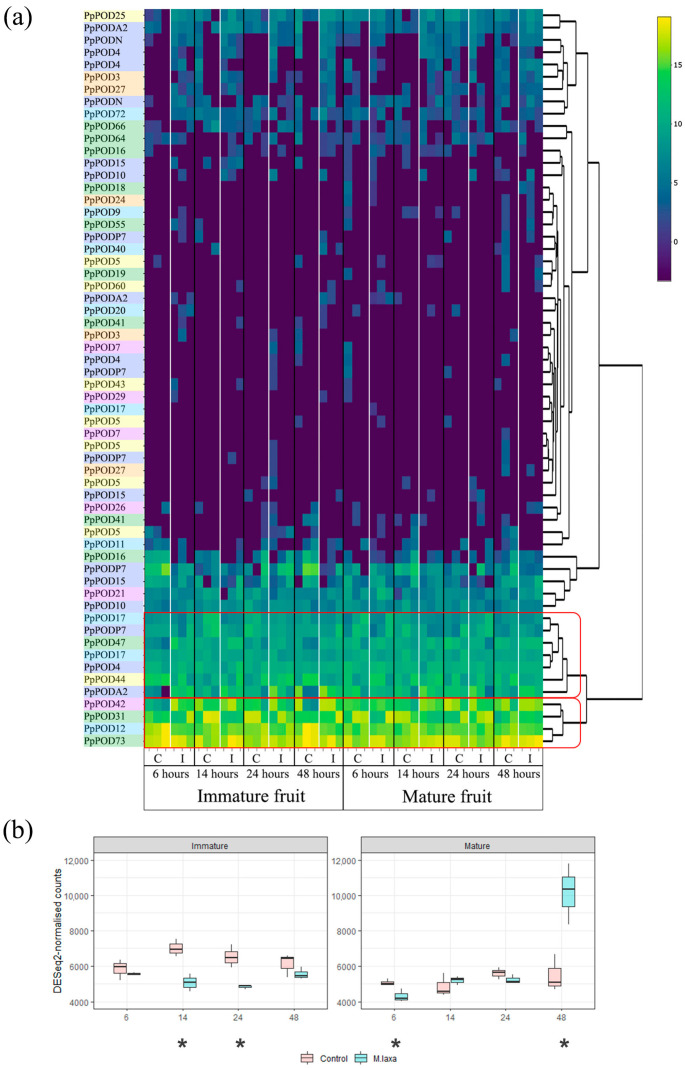
Expression pattern of the *POD* genes in healthy and infected with *M. laxa* peach (cultivar “Venus”; immature and mature fruits). (**a**) Expression of each *POD* gene of healthy (C) and infected (I) plants. The log2-based DeSEQ2-normalised counts were applied to build the heatmap. *POD* genes from subgroup A are marked with green boxes, subgroup B with orange boxes, subgroup C with yellow boxes, subgroup D with light blue boxes, subgroup E with purple boxes, and subgroup F with pink boxes. (**b**) The summary expression of all genes encoding peroxidase at 6, 14, 24, and 48 h after inoculation with fungus *M. laxa*. The data are presented as mean ± SE. A single asterisk on the bar represents a significant difference at * *p* < 0.05.

**Figure 6 plants-13-00127-f006:**
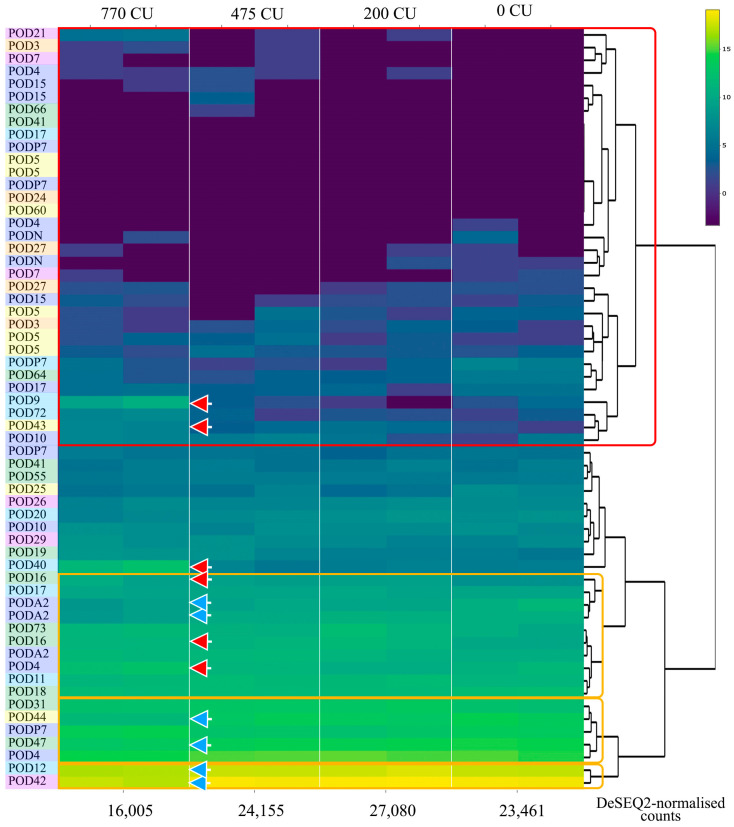
Expression pattern of the *POD* genes in peach under cold stress (cultivar “Fantasia”; flower buds). The log10-based DeSEQ2-normalised counts were applied to build the heatmap. *POD* genes from subgroup A are marked with green boxes, subgroup B with orange boxes, subgroup C with yellow boxes, subgroup D with light blue boxes, subgroup E with purple boxes, and subgroup F with pink boxes. Red and orange boxes denote low-expressed and high-expressed gene clusters, respectively. Up-regulated genes under cold stress are marked with a red arrow, and down-regulated genes are marked with a blue arrow (comparison between control and cold stress, Pajv < 0.01). The summary expression level is represented below the heatmap. CU—chilling unit.

**Table 1 plants-13-00127-t001:** Primer sequence of target and reference genes for RT-qPCR.

Gene Name	Forward Primer (5′-3′)	Reverse Primer (5′-3′)	Reference
*ACT*	GTTATTCTTCATCGGCGTCTTCG	CTTCACCATTCCAGTTCCATTGTC	[73]
*GAPDH*	ATTTGGAATCGTTGAGGGTCTTATG	AATGATGTTGAAGGAAGCAGCAC	[73]
*PODA2*	ACTTAGACCCCACAACTCCG	CTCCCCATTACTTCCCACCA	[25]
*POD4*	CATTGCTGCTCGAGACTCCGTT	AGCTGGCTGAGGGTAGAAGTGG	Present study
*POD5*	CAAGGGSTGCGATGCCTCAATT	CTGKCCTTGATCTCATCAATGA	Present study
*POD15*	TCCAGGGTTGTGATGGTTCG	AGACAACACCAGGGCAAACA	[24]
*POD31*	CCCTACTACAACGTACCGCT	ACCTGGATGAGCTGAGACAC	[25]

## Data Availability

Publicly available datasets were analysed in this study. These data can be found here: https://www.ncbi.nlm.nih.gov/sra/ accessed on 12 December 2023. The accession numbers of the SRA data are provided in this article. All the experimental data are contained within this article.

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
