# Peer review of "Class III Peroxidases in the Peach (Prunus persica): Genome-Wide Identification and Functional Analysis"

_plants, 2024, doi:10.3390/plants13010127_

Round 1

Reviewer 1 Report

Comments and Suggestions for Authors

The work is well written and easy to read, the experimental design is clear and the results good, the discussions are consistent with the results obtained. The figures are clear and well readable. The conclusion section could be improved by broadening future perspectives. The work can practically be published in the present form, I only point out a few minor details:

L47 scientific name

L54 add references

L60 justify this statement

L73 appears too informal, rewrite

L 412 what is meant by 'good'

L 440 add reference

L 447 add reference

Author Response

We thank the Reviewer very much for the careful reading of our manuscript and for providing useful comments.

We revised the manuscript in accordance with the comments of the Reviewer. Our comments and answers are given below.

REV: The conclusion section could be improved by broadening future perspectives.

Author: We rewritten Conclusions according to the Reviewer suggestion.

REV: The work can practically be published in the present form, I only point out a few minor details: L47 scientific name.

Author: We did not understand what the reviewer meant. Betula pendula is the scientific name for birch. In this case, the generic name is abbreviated as this is the second mention. The first record is in line 40.

REV: L54 add references and L60 justify this statement

Author: Thank you to the Reviewer for this comment. We have rewritten these sentences to make it clear that they refer to the same study - reference 16. Moreover, we have amended and merged into one paragraph the references to the papers analysing transgenic lines with increased expression of peroxidase genes. 

REV: L73 appears too informal, rewrite

Author: We’ve done.

REV: L 412 what is meant by 'good'

Author: We’ve written.

REV: L 440 and L 447 add reference

Author: We’ve done.

Reviewer 2 Report

Comments and Suggestions for Authors

The manuscript “Class III Peroxidases in Peach (Prunus persica): Genome-Wide 2 Identification and Functional Analysis” submitted by Vodiasova et al., was carefully reviewed. Peroxidase (POD) is a group of well-known large multi-gene family and that are broadly dispersed in living organisms. Their studies identified Peroxidases in Peach (Prunus persica) genome and performed a comprehensive bioinformatics analysis. qPCR and RNA-seq data also were employed to detect the expression level of candidate genes in tissue-specific and chilling stress.

However, peach as a common species in the Prunus genus, similar studies have been analyzed in detail in other species recent years. Such as https://doi.org/10.3390/ijms20112730, 10.1186/s12864-020-06828-z, https://doi.org/10.1186/s12864-021-07622-1,  In addition, POD gene expansion has also been systematically elaborated, and stress resistance and other traits have also been confirmed.

Moreover, it is a pity that the key genes screened have not been further verified, such as stress analysis of transgenic Arabidopsis or Saccharomyces cerevisiae is necessary.

Comments on the Quality of English Language

No.

Author Response

We thank the Reviewer very much for the careful reading of our manuscript and for providing useful comments.

We revised the manuscript in accordance with the comments of the Reviewer.

Our comments and answers are given below.

REV: However, peach as a common species in the Prunus genus, similar studies have been analyzed in detail in other species recent years. Such as https://doi.org/10.3390/ijms20112730, 10.1186/s12864-020-06828-z, https://doi.org/10.1186/s12864-021-07622-1,  In addition, POD gene expansion has also been systematically elaborated, and stress resistance and other traits have also been confirmed.

Author: The Reviewer is completely right that similar studies have been analysed in detail in other species recent years. However, such studies are becoming more and more numerous every year due to the need to study the functionality of each peroxidase gene for different species. The number of peroxidase genes is not the same for each species, and the functions of homologous genes may also vary. It is not completely understood what the different expression patterns of peroxidases under different stressors or in different tissues are related to. And accumulation of data to analyse expression patterns in different plants is necessary. Moreover, we presented analyses on all transcriptome data and showed that the expression of the same gene may differ among different genotypes of the same species (Results section). Therefore, we believe that this study, which presents preliminary data on the analysis of peroxidase genes in peach is relevant and timely. It should also be noted that such a detailed description of peroxidase genes for this species has been done for the first time.

REV: Moreover, it is a pity that the key genes screened have not been further verified, such as stress analysis of transgenic Arabidopsis or Saccharomyces cerevisiae is necessary.

Author: We thank the Reviewer for this comment and fully agree with the Reviewer's comment. We are planning to carry out such studies in the near future. The aim of this work was to conduct bioinformatic analyses that can serve as a basis for further studies, including those using transgenic Arabidopsis. We have rewritten more accurately the Discussion and Conclusion sections.

Reviewer 3 Report

Comments and Suggestions for Authors

Overall, this work was very descriptive and did not provide significant biological evidence. I would expect either to provide experimental data on the expression under cold stress and the related phenotypes or to access with biochemical approaches. 

Generally, the gene should be written in italics, and a professional editor should check the writing.

Comments on the Quality of English Language

A professional editor should check the writing.

Author Response

We thank the Reviewer very much for the careful reading of our manuscript and for providing useful comments.

We revised the manuscript in accordance with the comments of the Reviewer.

Our comments and answers are given below.

REV: Overall, this work was very descriptive and did not provide significant biological evidence. I would expect either to provide experimental data on the expression under cold stress and the related phenotypes or to access with biochemical approaches. 

Author: According to the Reviewer suggestion we have provided all analysed transcriptomic data and calculated DeSEQ-normalized counts for each POD genes in Supplementaries. Moreover, we added the description of related phenotypes for each analysis. Unfortunately, all experimental data (including those available at NCBI) analysed in this paper are not accompanied by biochemical methods to analyse the level of peroxidase activity. Nevertheless, based on literature data, it is known that both fungal infection and cold stress lead to an increase in the activity of this enzyme. As for the analysis of peroxidase activity in different varieties, we plan to conduct such studies in the future (the collection of the Nikita Botanical Garden currently contains more than 150 different peach varieties).

REV: Generally, the gene should be written in italics, and a professional editor should check the writing.

Author: We have rewritten all gene name in italics. Also, as the reviewer recommended, we checked and corrected the text with English speaker.

Reviewer 4 Report

Comments and Suggestions for Authors

1) In this study, the RNA-seq data obtained from NCBI were found to be incomplete, featuring only 1-2 biological replicates for analysis, but the original data have four biological replicates. Consequently, the transcriptome analysis results demonstrated poor reliability. Typically, transcriptome data necessitate further validation through qPCR, however, a significant portion of the data remains unverified, thereby compromising the overall reliability of the conclusions drawn.

2) This study undertook an analysis of the POD gene family within Prunus persica across diverse tissues and varieties. However, the findings predominantly highlight the high and low expression levels of PODs without providing a comprehensive identification of specifically or differentially expressed PODs in distinct tissues. Additionally, the study lacks a comparative assessment of whether these differential expressions remain consistent across various varieties. This limitation is also evident in the analysis of responses to cold stress and disease resistance. Therefore, a more in-depth data analysis is needed to elucidate the specific and differential expression patterns of PODs in different tissues, varieties, and stress.

3) Noteworthy conceptual and graphical discrepancies were identified within the manuscript. For instance, the peroxidase EC number is correctly identified as 1.11.1.x, with Class III peroxidase (EC number 1.11.1.7) representing a subset of peroxidases. However, there is a degree of confusion in the article regarding these concepts. Additionally, Figure 4's content deviates from the descriptions provided in the text, thereby introducing inconsistencies in the presentation of data.

Author Response

We thank the Reviewer very much for the careful reading of our manuscript and for providing useful comments.

We revised the manuscript in accordance with the comments of the Reviewer. Moreover, we have recalculated some data and rewritten Discussion and Conclusion. Our comments and answers are given below.

REV: In this study, the RNA-seq data obtained from NCBI were found to be incomplete, featuring only 1-2 biological replicates for analysis, but the original data have four biological replicates. Consequently, the transcriptome analysis results demonstrated poor reliability. Typically, transcriptome data necessitate further validation through qPCR, however, a significant portion of the data remains unverified, thereby compromising the overall reliability of the conclusions drawn.

Author: We are grateful to the Reviewer for this helpful remark and fully agree with all of it. We redid the entire bioinformatic analysis of the transcriptome by including absolutely all libraries available for each case. As a result, we found some inaccuracies and changed some conclusions. We also agree that transcriptome data should be validated with qPCR data. Unfortunately, this is not possible, as the analyses that were done for different cultivars showed that the gene expression pattern may differ from cultivar to cultivar (which is probably associated with different stress tolerance). And hence, validation should be performed on the same cultivar on which the transcriptome analysis was done. However, we supplemented our analysis by calculating differential expression using the DeSEQ method, and separately identified genes that are significantly different according to the analysis. It should be noted that DeSEQ is stringent to detect outliers and excludes genes with extreme read counts by default and the false positive rate for DEG is 0% at adjusted P values less than 0.05 (Anders et al. 2013; Rajkumar et al. 2015). The results of the analyses are presented in the Supplementary Material.

REV: This study undertook an analysis of the POD gene family within Prunus persica across diverse tissues and varieties. However, the findings predominantly highlight the high and low expression levels of PODs without providing a comprehensive identification of specifically or differentially expressed PODs in distinct tissues. Additionally, the study lacks a comparative assessment of whether these differential expressions remain consistent across various varieties. This limitation is also evident in the analysis of responses to cold stress and disease resistance. Therefore, a more in-depth data analysis is needed to elucidate the specific and differential expression patterns of PODs in different tissues, varieties, and stress.

Author: Thank you for this comment. In response to the previous comment, we indicated that we supplemented the study with a deeper analysis of peroxidase gene expression and identified genes that showed differential expression with adjusted P values less than 0.01. Such analysis was performed for different tissues, cultivars, and stress.

REV: Noteworthy conceptual and graphical discrepancies were identified within the manuscript. For instance, the peroxidase EC number is correctly identified as 1.11.1.x, with Class III peroxidase (EC number 1.11.1.7) representing a subset of peroxidases. However, there is a degree of confusion in the article regarding these concepts. Additionally, Figure 4's content deviates from the descriptions provided in the text, thereby introducing inconsistencies in the presentation of data.

Author: We fully agree with the reviewer. In this study, we only analyse Class III peroxidase. We have also corrected the description to Figure 4.

Round 2

Reviewer 2 Report

Comments and Suggestions for Authors

The author has carefully revised the concerns I proposed and it is currently meets the publishing requirements.

Reviewer 3 Report

Comments and Suggestions for Authors

The authors have addressed my previous comments.